# Transcriptome Co-Expression Network Analysis of Peach Fruit with Different Sugar Concentrations Reveals Key Regulators in Sugar Metabolism Involved in Cold Tolerance

**DOI:** 10.3390/foods12112244

**Published:** 2023-06-01

**Authors:** Lufan Wang, Xiaolin Zheng, Zhengwen Ye, Mingshen Su, Xianan Zhang, Jihong Du, Xiongwei Li, Huijuan Zhou, Chen Huan

**Affiliations:** 1Forestry and Fruit Research Institute, Shanghai Academy of Agricultural Sciences, Shanghai 201403, China; luf0817@163.com (L.W.); yezhengwen1300@163.com (Z.Y.); sumingshen@saas.sh.cn (M.S.); z.xn2009@163.com (X.Z.); jihonghb@126.com (J.D.); lixiongweisea@163.com (X.L.); 2College of Food Science and Biotechnology, Zhejiang Gongshang University, Hangzhou 310018, China; zheng9393@163.com; 3Jiangsu Key Laboratory for Horticultural Crop Genetic Improvement, Nanjing 210000, China

**Keywords:** chilling injury, peach fruit, regulatory network, sugar metabolism, transcriptome analysis

## Abstract

Peach fruits are known to be highly susceptible to chilling injury (CI) during low-temperature storage, which has been linked to the level of sugar concentration in the fruit. In order to better understand the relationship between sugar metabolism and CI, we conducted a study examining the concentration of sucrose, fructose, and glucose in peach fruit with different sugar concentrations and examined their relationship with CI. Through transcriptome sequencing, we screened the functional genes and transcription factors (TFs) involved in the sugar metabolism pathway that may cause CI in peach fruit. Our results identified five key functional genes (*PpSS*, *PpINV*, *PpMGAM*, *PpFRK*, and *PpHXK*) and eight TFs (*PpMYB1/3*, *PpMYB-related1*, *PpWRKY4*, *PpbZIP1/2/3*, and *PpbHLH2*) that are associated with sugar metabolism and CI development. The analysis of co-expression network mapping and binding site prediction identified the most likely associations between these TFs and functional genes. This study provides insights into the metabolic and molecular mechanisms regulating sugar changes in peach fruit with different sugar concentrations and presents potential targets for breeding high-sugar and cold-tolerant peach varieties.

## 1. Introduction

Peach (*Prunus persica* (L.)) is extensively cultivated in China, ranging from 23° to 45° N. China is the leading global producer of peach fruit [1], which is renowned for its nutritional value and distinctive flavor. However, peach fruit is susceptible to chilling injury (CI) during low-temperature storage, which compromises its flavor and quality, thereby failing to satisfy consumer preferences [2].

In fruit, the quality and flavor are predominantly determined by the concentration and composition of sugars. Soluble sugars, including sucrose, fructose, and glucose, accumulate in the fruit and are important for growth and development, serving as energy substances and providing resistance to external abiotic stresses [3]. Changes in sugar concentrations in leaves and roots can significantly affect photosynthesis and respiratory metabolism, respectively [4]. Environmental stresses, such as temperature stress, drought, and salt, can limit plant growth and development and cause significant crop losses [1,5]. Among the various stresses, low temperature stress holds significant importance [6]. In the case of postharvest fruits, low temperature stress primarily arises from storage under low temperature conditions, a prevalent practice in postharvest fruit preservation. However, it is unfortunate that peach fruit, similar to other tropical fruits, exhibit sensitivity towards low temperatures. This sensitivity often leads to profound physiological and biochemical alterations, manifesting as symptoms of CI [7,8]. To mitigate the effects of stress, plants adjust their soluble sugar concentrations, such as through the acceleration of sucrose decomposition in response to drought [9], and the accumulation of soluble sugars in vacuoles to improve salt resistance [10]. In postharvest fruits, low-temperature storage can improve cold resistance by inducing soluble sugar changes, as demonstrated by sucrose accumulation in cloudberry and higher sucrose levels stabilizing membrane permeability in peach [2,11]. In summary, plants have abilities to modulate changes in soluble sugars in response to low-temperature stress, while sugar metabolism is in charge of modulating the concentration and proportion of soluble sugars during fruit development and ripening.

Sugar metabolism in postharvest fruit is a complex process that involves multiple pathways, mainly including sucrose metabolism, hexose metabolism, sorbitol metabolism, and starch metabolism. These pathways work together to maintain the normal ripening and senescence of fruits. Sucrose is constantly synthesized and broken down in the depot tissue to provide raw materials and energy for the fruit. Consequently, studies related to sucrose metabolism are of great importance in understanding the sugar accumulation process in peach fruit.

Soluble sugars act as cryoprotectants, active oxygen species, osmotic pressure regulators, and signaling molecules, and can delay the onset of fruit low-temperature stress [12]. In recent years, studies have explored the close relationship between sugar metabolism and cold resistance at the molecular level. Overexpression of *PpCBF6* can increase the sucrose level of peach fruit to improve its cold resistance via down-regulating the expression of *PpVIN2*, while *PpZAT10* can indirectly affect sucrose metabolism via inhibiting *PpVIN2* in peach fruit and enhancing VIN activity [13]. Despite progress in understanding the molecular mechanisms underlying sugar metabolism and cold resistance, there are still some gaps in our knowledge. Specifically, there are few reports on the differences in CI in peach fruit with different sugar concentrations of the same variety, and it is unclear which genes play a critical role in regulating the relationship between sugar metabolism and CI.

Transcriptomics is a powerful approach for analyzing gene function and structure at a holistic level and has been widely used for candidate gene discovery, functional identification, and genetic improvement in plants [14]. RNA-seq technology, enabled by next-generation sequencing platforms, has become an important tool for investigating molecular regulatory mechanisms in postharvest fruits [15]. Transcriptome analysis has revealed regulatory networks involved in low-temperature stress, high-sucrose accumulation, and anthocyanin synthesis in various fruit species [16,17,18]. Transcription factors (TFs), such as ERF [19], MYB [20], bZIP [21], and WRKY [22], have been implicated in sugar metabolism and low-temperature stress. However, no transcriptome analysis has yet been reported on comparing the expression patterns of sugar metabolism-related genes in peach fruit with different sugar concentrations during cold storage. In this study, based on bioinformatics analysis, transcriptomic analysis, and weighted gene co-expression network analysis (WGCNA), the key candidate genes (*PpSS*, *PpINV*, *PpMGAM*, *PpFRK,* and *PpHXK*) and TFs (*PpMYB1/3*, *PpMYB-related1*, *PpWRKY4*, *PpbZIP1/2/3,* and *PpbHLH2*) involved in sugar metabolism were identified. This study may provide a valuable reference for understanding the mechanisms underlying the relationships between the CI development and sugar accumulation in peach fruit.

## 2. Materials and Methods

### 2.1. Plant Material

Approximately 5000 “Hujingmilu” peach fruit at 70–80% maturation (at about 110–115 days after full blooming, a peach fruit’s external color changes from green to cream, with firmness ranging from 16 to 18 N, Appendix A) were harvested from the orchard in the Pudong District, Shanghai, China. The fruit selection was based on total soluble solid (TSS) concentration, and 3 groups were created with approximately 200 fruit in each group for 3 replicates: high−sugar group (H group, TSS greater than 14%), middle−sugar group (M group, TSS between 12–14%), and low−sugar group (L group, TSS less than 12%). The TSS concentration of peach fruit at harvest was determined using a non-destructive fruit brix tester (H100F, Beijing, China, Appendix A). After grouping, all peach fruit were stored at 4 ± 0.5 °C and 85–90% relative humidity for 4 weeks. Peach flesh was sampled at 0, 21, and 28 d, flash-frozen in liquid nitrogen, and stored at −80 °C for further laboratory analysis. At 21 d and 28 d, some fruits were, respectively, moved to room temperature (20 ± 1 °C) for 5 days to evaluate the occurrence of CI.

### 2.2. Measurement of CI Index

The extent of CI was determined by a CI index, and the CI scale was divided into 4 grades, where 0 indicated no flesh area affected, 1 indicated 0–1/4 flesh area affected, 2 indicated 1/4–1/2 flesh area affected, 3 indicated 1/2–3/4 flesh area affected, and 4 indicated 3/4–1 flesh area affected. The CI index was evaluated according to our previous study [23].

### 2.3. Measurement of Firmness, TSS and Sugar Concentration during Storage

The flesh firmness of peach was determined using a Using texture analyzer (TA. XT. Plus, SMS, CA, British) and a cylindrical probe (P/5) with a diameter of 5 mm. A two deformation test (TDT) was used to press down twice, with a pre-test speed of 60 mm min^−1^, a test speed of 120 mm min^−1^, a post-test speed of 600 mm min^−1^, and a trigger force of 5 g.

The clarified peach juice was filtered by centrifuge and was determined by a Pocket Brix-Acidity Meter (PAL-BX/ACID 8, Atago, Saitama, Japan). The concentration of sucrose, fructose, and glucose was determined as Zhou et al. [24] described and were expressed as g kg^−1^ fresh weight.

### 2.4. RNA Sequencing

#### 2.4.1. RNA Extraction, Library Construction, and Sequencing

Our findings showed that high− and low−sugar peach fruit had the most significant differences in both CI index and sugar concentration among the three groups in peach fruit during storage. Therefore, six samples (H 0/21/28 d and L 0/21/28 d) from these two groups were used for sequencing with three biological replicates.

Total RNA was extracted from frozen materials (peach flesh) using the plant RNA Extraction Kit (TIANGEN, Beijing, China) and was used for cDNA library construction.

Transcriptomic sequencing was completed by Magi Biomedical Technology Co., Ltd. (Shanghai, China). Briefly, high−quality RNA of three biological replicates was processed using the Illumina TruseqTM RNA sample preparation Kit for library construction. mRNA was separated from the total RNA by A-T base pairing with ploy-A using magnetic beads with Oligo (dT). mRNA was fragmented into fragments of about 300 bp by adding fragmentation buffer under appropriate conditions. The cDNA was synthesized by reverse transcription, then the sorted products were amplified by PCR and the final library was purified. Finally, quantification was performed by the QuantiFluor^®^ dsDNA System, and on-machine sequencing was performed by the Illumina HiSeq xten/NovaSeq 6000 sequencing platform.

#### 2.4.2. Differentially Expressed Genes (DEGs) Analysis

The gene expression levels were analyzed using the fragments per kilobase million (FPKM) method for each sample. DESeq2 software was used to analyze the differences between 2 different groups, DEGs were chosen with log_2_|Fold Change| ≥ 1 and false discovery rate (FDR) parameter < 0.05.

Principal component analysis (PCA) was used to exclude outlier samples, and the Pearson correlation coefficient was performed to reflect the correlation between samples. Gene Ontology (GO) was used to analyze DEGs annotation based on molecular functions (MF), cellular components (CC), and biological processes (BP). The *Kyoto Encyclopedia of Genes and Genomes pathway* (KEGG) databases were used to identify the significantly enriched pathways among DEGs [25].

#### 2.4.3. WGCNA Analysis and Gene Network Construction

WGCNA was used to identify genes with similar expression patterns and to analyze the correlation between module and sample phenotypes. It was performed using a WGCNA R package (v1.49) [26]. After filtering, 3920 genes were screened and their expression values were imported into WGCNA. The automatic network construction function block-wise modules were used to advance the construction of the co-expression modules. In the WGCNA network, the soft thresholding was 9, the merge cut height was 0.25, and the min Module Size was 30 [27,28]. The hub gene related to sucrose, fructose, and glucose concentration in the “MEpink” module was used to construct co-expression networks and visualized the candidate genes using the Cytoscape (v3.9.1) software [29]. The putative cis-elements in the upstream 2000 bp promoter sequence of the candidate genes were performed using the *Plantcare* database (http://bioinformatics.psb.ugent.be/webtools/plantcare/html/, accessed on 11 January 2023). 

#### 2.4.4. qRT-PCR Analysis

To validate the reliability of the RNA-seq results, 13 genes were chosen for qRT-PCR analysis according to our previous study [23]. The translation elongation factor 2 (TEF2) was used as the internal reference gene [30], and the primers were shown in Appendix A.

### 2.5. Statistical Analysis

All statistical analyses were analyzed by Microsoft Excel (Microsoft, Redmond, WA, USA) and SPSS (SPSS Inc., Chicago, IL, USA), and Duncan’s multiple comparison method was used for the significance analysis of data (*p* < 0.05).

## 3. Results

### 3.1. Sugar Concentration Was Associated with CI Development

As shown in Figure 1A, the CI symptom occurred in peach fruit at 21 d, with no difference in CI index observed in the different sugar groups. At 28 d, the CI index in low−sugar peach fruit was 2.10 and 1.27 times higher than in high−sugar and middle-sugar peach fruit, respectively. Figure 1B illustrated a decreasing trend in TSS in high−sugar peach fruit from 0 d to 28 d. Conversely, the TSS in middle− and low−sugar peach fruit increased initially and then decreased, with TSS being the highest in high−sugar peach fruit. Figure 1C shows that the sucrose concentration decreased in all fruit, but high-sugar peach fruit had a higher sucrose concentration than low-sugar peach fruit. The fructose concentration in high− and middle−sugar peach fruit remained stable during storage, but decreased in low−sugar peach fruit from 21 d to 28 d (Figure 1D). There was no significant difference in fructose concentration between high− and middle−sugar peach fruit during the whole storage, but both had higher concentrations than low−sugar fruit. The glucose concentration in all fruit did not change significantly, but high− and middle−sugar peach fruit had higher glucose concentrations than low−sugar peach fruit (Figure 1E). These results indicated that the sugar concentration in peach fruit was negatively correlated with the degree of CI development.

### 3.2. Transcriptome Analysis in Peach during Cold Storage

To investigate the differences in gene expression between high−sugar and low−sugar peach fruit, transcriptomic analysis was performed on six samples. For each sample, 6.3 Gb of raw data and at least 6.1 Gb of clean data were obtained, as well as at least 4.15 × 10^7^ clean reads. A high percentage of the clean reads (95.56%) could be mapped to the reference genome, and both the Q20 and Q30 values were above the acceptable threshold of 97.81% and 93.64% (Appendix A), respectively, indicating the quality of the library construction and sequencing was sufficient for further analysis. The expression distribution of the six sample groups was visualized using a violin plot, with the gene expression levels shown to be evenly distributed and well−enriched across the samples (Figure 2A). A Pearson’s correlation analysis was performed to assess the reliability of the biological replicates in the transcriptome data, revealing highly correlated characteristics (R^2^ > 0.97) among the 3 replicates within each sample (Figure 2B and Appendix A). Based on the gene expression data for all samples, the PCA results showed that the 3 principal components—PC1, PC2, and PC4—explained 73.22%, 11.78%, and 3.45% of the between-group variation, respectively. The results of the PCA showed that the high- and low-sugar fruit samples were not spatially clustered in distinct groups, indicating a significant between-group variation (Figure 2C). However, the biological replicates within each sample were clustered together, demonstrating good reproducibility.

### 3.3. Comprehensive Analysis of DEGs

The analysis revealed that 1875, 1142, and 2001 DEGs were present in the comparisons between low−sugar (L) and high−sugar (H) peach fruit samples at 0, 21, and 28 d (Figure 2D), respectively. The analysis also showed that 3920 DEGs were present across the 3 comparisons (L 0 vs. H 0, L 21 vs. H 21, and L 28 vs. H 28). Additionally, 177 DEGs were found across all 3 comparison groups; 217, 162, and 265 DEGs were found in 2 comparison groups; and 1216, 1297, and 486 DEGs were found in 1 comparison group (Figure 2E), respectively.

The DEGs were annotated using the GO database. After annotation, the top 20 functional groups annotated in the comparisons between L 0 and H 0, L 21 and H 21, and L 28 and H 28 included 5 groups in MF, 8 groups in BP, and 7 groups in CC (Appendix A), respectively.

Furthermore, a comparison of the DEGs with the KEGG database was performed to reveal the enriched pathways. The top 20 pathways enriched for DEGs in each comparison are, respectively, shown in Figure 3A–C. Among these pathways, starch and sucrose metabolism were significantly enriched in the three comparisons.

### 3.4. Identification of Sugar-Related DEGs Co-Expression Modules by WGCNA

To examine the transcriptional regulation of sugar metabolism in peach fruit, we conducted WGCNA on 3920 DEGs that were differentially expressed in response to changes in sucrose, fructose, and glucose concentrations (Figure 4B). The analysis resulted in the classification of the DEGs into 11 distinct modules (Figure 4C). The correlation analysis of the modules and sugar traits revealed that the MEpink and MEblack modules were significantly correlated with changes in sucrose concentration. The results showed a positive correlation between gene expression and sucrose concentration in the MEpink module, with a coefficient of 0.72 (*p* = 8 × 10^−4^). Conversely, in the MEblack module, gene expression was negatively correlated with sucrose concentration, with a coefficient of −0.45 (*p* = 0.06). Given the strong positive correlation between the MEpink module and sucrose concentration, we further focused on the MEpink module and identified a total of 1468 genes that were associated with changes in sucrose concentration.

### 3.5. Identification of Functional Genes Related to Sugar Metabolism

The results of GO annotation of 1468 DEGs that were positively related to sucrose concentration as revealed by WGCNA were analyzed. These DEGs were grouped into 38 functional pathways, including 16 in BP, 11 in CC, and 11 in MF (Appendix A).

In addition, 1468 DEGs were assigned to 104 pathways based on KEGG pathway enrichment analysis (Figure 5). Notably, starch and sucrose metabolic pathways were significantly enriched among the top 20 enriched pathways in the MEpink module. Five genes were identified in the pathway, including *PpSS* (Sucrose synthase, Prupe.5G241700), *PpMGAM* (alpha-glucosidase, Prupe.4G103600), *PpINV* (Invertase, Prupe.3G009500), *PpFRK* (Fructokinase, Prupe.3G160500), and *PpHXK* (Hexokinase, Prupe.3G057800) (Its related information is shown in Appendix A). A heatmap created using TBtools showed that *PpHXK* and *PpFRK* were up-regulated (Figure 6), indicating a faster rate of hexose phosphorylation in high-sugar peach fruit than in low-sugar peach fruit during cold storage. Conversely, *PpMGAM* and *PpINV* were down-regulated, indicating a slower rate of sucrose degradation in high-sugar peach fruit in the late storage period.

### 3.6. Expression Analysis of TFs Involved in Sugar Metabolism at Low Temperature

To gain further insights into the transcriptional regulation mechanisms of sugar metabolism, we performed a TF prediction analysis on the 1468 DEGs obtained from the WGCNA analysis. The results revealed a total of 81 TFs (Figure 7) and its related information is shown in Appendix A. Of these, the top 10 families of TFs were the ethylene responsive factor (ERF), HB−other, myeloblastosis (MYB), N−acetylcys−teine (NAC), WRKY, basic helix-loop-helix (bHLH), basic leucine-zipper (bZIP), auxin response factors (ARF), heat shock factor (HSF), and MYB−related, with 8, 7, 7, 6, 5, 5, 4, 3, 3 and 2 members, respectively. The abundance of each TF at each sampling site was depicted in Figure 7, and it was found that the transcription levels of the TFs in the high-sugar peach were significantly higher compared to the low−sugar peach at 0 d and 28 d.

### 3.7. Regulatory Network of Functional Genes and TFs

To uncover the transcriptional regulation mechanism of sugar metabolism in peach fruit, a co-expression network was constructed using 5 functional genes and 81 TFs screened above (Figure 8). The results showed that these functional genes were regulated by multiple TFs, which indicated the complexity of the regulation of sugar metabolism in peach fruit. Since TFs and functional genes had higher correlation coefficients in the inner circle, the top ten TF families in the inner circle of the network−including ERF (*PpERF2/3*), HB-other (*PpHB-other1*/2/3), MYB (*PpMYB1/3/6*), NAC (*PpNAC1/4*), WRKY (*PpWRKY4*), bHLH (*PpbHLH1/2/5*), bZIP (*PpbZIP1/2/3*), ARF (*PpARF1/3*), HSF (*PpHSF1/2/3*), and MYB−related (*PpMYB-related1*)−were selected for subsequent experiments.

### 3.8. Cis-Component Predictive Analysis

There were some false positives because the co-expression network was only linked based on gene expression. To reduce the probability of false positive results, we used the *Plantcare* database to analyze the cis-acting elements in the promoter regions of the five functional genes. The results, presented in Figure 9, showed that all five functional genes contained common cis-acting elements (TATA−box and CAAT−box). Other cis-acting elements were primarily associated with stress response (W box), drought induction (MYC), abscisic acid response (ABRE), jasmonic acid response (CGTCA−motif and TGACG-motif), auxin response (TGA−element), and light response (MYB, Box 4 and G−box). Based on these findings, we selected eight TFs that might have regulatory roles in response to abiotic stresses for further analysis, including *PpMYB1* (Prupe.6G229000), *PpMYB3* (Prupe.5G182000), *PpMYB-related1* (Prupe.2G176200), *PpWRKY4* (Prupe.8G265900), *PpbZIP1* (Prupe.7G160600), *PpbZIP2* (Prupe.8G091600), *PpbZIP3* (Prupe.1G374400), and *PpbHLH2* (Prupe.6G159200).

### 3.9. Verification of DEGs by qRT−PCR

To assess the reliability of the RNA−Seq DEG data, five functional genes and eight TFs were selected for quantification using qRT−PCR. As shown in Figure 10, the expression patterns of *PpSS*, *PpMGAM*, *PpINV*, *PpFRK,* and *PpHXK* were consistent over the entire storage period. *PpMGAM* and *PpINV* were key enzymes involved in sucrose hydrolysis to generate hexose. These two genes in high− and low−sugar peach fruit exhibited a declining pattern from 0 d to 28 d. Notably, the expression levels of *PpMGAM* and *PpINV* in high−sugar peach fruit were relatively lower at day 21 compared to low-sugar peach fruit, indicating a lower rate of sucrose degradation. The gene expression of *PpSS* was higher in high-sugar peach fruit than in low−sugar peach fruit at 0 d and 28 d, suggesting its role in regulating sucrose synthesis. *PpFRK* and *PpHXK*, which were mainly involved in hexose phosphorylation, were significantly decreased in high-sugar peach fruit compared to low−sugar peach fruit at 21 d. *PpbZIP3* demonstrated a significant increase in expression levels at 21 d in low−sugar peach fruit. Conversely, the expression levels of *PpMYB1*, *PpWRKY4*, *PpMYB-related1*, and *PpbZIP1* increased progressively at 21 d and 28 d. Meanwhile, the expression pattern of *PpbHLH2* was similar to that of *PpSS*, *PpFRK*, and *PpHXK*, which displayed a trend of initial decline at 21 d, but an upward trend at 28 d. Overall, the qRT-PCR results provided further validation of the RNA−Seq DEG data.

## 4. Discussion

The utilization of low temperatures has been established to delay fruit ripening and extend its shelf life. However, some crops, such as banana [31], pear [32], and peach [33]—which are cold−sensitive— are highly susceptible to low temperatures. Our results showed that peach fruit in all three groups displayed CI symptoms at 21 d of storage (Figure 1A). However, the CI index was found to be lower in the high-sugar peach fruit at 28 d, suggesting that a higher sugar concentration can delay the development of CI symptoms, which is consistent with previous studies [34]. The physiological and biochemical responses are utilized by plants to improve their resistance to low-temperature stress, such as the accumulation of soluble sugars. Soluble sugar protects cells from the damage caused by the low-temperature stress through several mechanisms. Firstly, the accumulation of soluble sugar reduces the freezing point and enhances the water retention capacity of cells, thereby preventing ice crystal formation. Secondly, the sugar metabolism process generates energy and other protective substances, which help in maintaining cellular functions under stress. Thirdly, soluble sugars provide a protective role in preserving the cellular material and biofilm. Increasing the sugar concentration of postharvest fruits has been shown to improve their cold resistance [35,36]. The soluble sugars of ‘hujingmilu’ peach fruit were primarily sucrose (Figure 1C), and the high-sugar peach fruit with a higher sucrose concentration showed a lower CI index (Figure 1A), indicating a negative correlation between the accumulation of sucrose and the CI index. A high sucrose concentration that can prevent the development of CI symptoms was reported by Abidi et al. [34]. Additionally, our previous experimental results demonstrated a positive correlation between sucrose accumulation and cold tolerance [23]. These results further confirmed that the accumulation of sucrose was positively correlated with cold tolerance. Unlike sucrose, fructose and glucose were found to be present in lower concentrations, but their changing trends remain the same during storage (Figure 1D,E). Previous studies have shown that glucose and fructose followed a similar pattern during fruit development, and their concentrations remain consistent at each time point [37]. In summary, our results indicated that the accumulation of soluble sugars in peach fruit was associated with a delay in CI, albeit with a low concentration of hexose observed throughout the storage period. We speculated that sucrose accumulation was the primary factor contributing to the enhanced cold resistance of peach fruit.

In this study, five functional genes, namely *PpSS*, *PpMGAM*, *PpINV*, *PpFRK,* and *PpHXK*, were positively correlated with sucrose concentration via WGCNA analysis (Figure 4C). The transport of sucrose from leaves to cells occurs via short-distance transport to the bast, followed by long-distance transport into the cell. Once inside the cell, sucrose is degraded by SS-c or INV and synthesized through the catalysis of SPS in the cytoplasm [38]. The *PpSS* expression was decreased in the high-sugar fruit at 21 d (Figure 10), suggesting a role in evaluating the composition of sugars, consistent with previous studies in tomato, watermelon, and melon [39,40,41]. *PpINV* showed a decreasing trend throughout the storage period, with higher gene expression levels in the high-sugar fruit compared to the low-sugar fruit (Figure 10). This may be due to an excessive sucrose accumulation, providing a primary substrate for the maintenance of internal fruit life activities, which is similar to our previous findings [23]. Because of its dual role in the catalytic reaction of sugars, MGAM has been more commonly reported for use as a drug. It is not only in animals but also in plants and microorganisms [42,43]. In our study, *PpMGAM* gene levels were down-regulated in high-sugar fruit during late storage, negatively correlating with sucrose concentration, suggesting a role in reducing the rate of sucrose degradation (Figure 10). However, further verification is needed to confirm its functionality. HXK and FRK are hexose phosphorylase enzymes that play an important role in the metabolism and distribution of library tissues in plants. They also act as signal sensors in plants, regulating metabolism and growth by influencing the plant life cycle [9]. The gene expression levels of *PpFRK* and *PpHXK* were significantly down-regulated in high-sugar fruit during the early stage of storage, delaying the phosphorylation of fructose and glucose and resulting in higher concentrations of these sugars (Figure 10). Conversely, the accumulation of fructose and glucose concentrations inhibited the expressions of *PpFRK* and *PpHXK* during the late storage period, which was consistent with those reported by Li et al. [44]. Our findings indicated that the expression of *PpFRK* and *PpHXK* was influenced not only by environmental factors, but also by the concentration of hexose. We speculated that the high concentration of soluble sugars in the high-sugar peach fruit is adequate to sustain its metabolic activities, thereby negating the requirement for substantial hexose phosphorylation to provide energy.

Biological processes require the regulation of TFs to form a complex regulatory network [45], with the expression of functional genes primarily regulated by their upstream TFs. In this study, through the co-expression network (Figure 8), we identified a total of 23 TFs that may regulate the functional genes under low-temperature stress, including *PpERF2/3*, PpHB-other1/2/3, *PpMYB1/3/6*, *PpNAC1/4*, *PpWRKY4*, *PpbHLH1/2/5*, *PpbZIP1/2/3*, *PpARF1/3*, *PpHSF1/2/3*, and *PpMYB-related1*. Previous studies have demonstrated the involvement of various TFs in sugar metabolism or resistance to low-temperature stress in plants. For example, ERF can recognize the GCC-box (AGCCGCC) element and then act as an activator or repressor of specific genes [46]. *OsERF2*, identified by Xiao et al. [47], was found to regulate the accumulation of sucrose and UDPG by modulating the expression of genes related to sucrose metabolism and hormone signaling pathways. In peach fruit, that the ERF gene family might play a critical role in regulating CI during the early stages of cold storage was founded by Muto et al. [48]. WRKYs can recognize the W-box element of target gene promoter sequences, and Li et al. [49] demonstrated that *PpWRKY40* could bind to W-box elements in the promoters of *PpSS1* and *PpSPS3*, activating their transcription. bZIPs are mainly involved in plant growth and development, biological and abiotic stress, and secondary metabolism regulation by recognizing G-box elements of functional genes. *PpbZIP23* and *PpbZIP25* were candidate genes for ABA-dependent cold signaling and reduced CI index under low-temperature stress conditions was reported by Aslam et al. [50]. Additionally, S1-like bZIP in “arabidopsis” was a known sucrose-specific signal transducer [51]. The MYB family TFs play an important role in regulating plant hormones and secondary metabolism, and that the gene expression of MYB-6 was up-regulated in carrots under low-temperature conditions was shown by Dar et al. [52]. Additionally, NAC was demonstrated to regulate resistance to low temperatures in “Elymus nutans” and “Capsicum annuum” [53]. The above studies indicated that TFs can respond to low-temperature stress by regulating target genes. However, it remains to be investigated whether the TFs identified in our study can regulate the five functional genes in response to low-temperature stress. Further research is needed to elucidate the regulatory mechanisms of TFs in the sugar metabolism pathway and their roles in the response to low-temperature stress in peach fruit.

TFs are essential for the regulation of gene expression by binding to specific cis-acting elements in gene promoter regions [54]. Analysis of the promoter regions indicated the presence of commonly occurring cis-acting elements, including MYB, W-box, ABRE, MYC, and G-box (Figure 9). Based on this analysis, eight TFs (*PpMYB1/3*, *PpMYB-related1*, *PpWRKY4*, *PpbZIP1/2/3*, and *PpbHLH2*) with the highest likelihood of regulation were selected for further analysis via qRT-PCR (Figure 10). Notably, the gene expression levels of multiple TFs followed the same trend as the functional genes, with a significant down-regulation observed in high-sugar fruit at 21 d, followed by a significant up-regulation at 28 d. This trend was positively correlated with the changes in sucrose levels. However, further research is needed to understand the precise network regulation and the response of these TFs to sugar signals in peach fruit.

## 5. Conclusions

In summary, this study used RNA-Seq data to analyze the effect of sugar concentration on the cold tolerance of peach fruit under low-temperature conditions. Our findings showed that peach fruit with high sugar concentration exhibited an increased cold tolerance, which was regulated by the transcription levels of genes and TFs involved in sugar metabolism. Specifically, we identified five key functional genes (*PpSS*, *PpINV*, *PpMGAM*, *PpFRK*, and *PpHXK*) and eight TFs (*PpMYB1/3*, *PpMYB-related1*, *PpWRKY4*, *PpbZIP1/2/3*, and *PpbHLH2*) that play important roles in regulating sugar metabolism and cold tolerance in peach fruit. These results provided valuable genetic resources for the development of new cold-tolerant peach fruit varieties through molecular breeding.

## Figures and Tables

**Figure 1 foods-12-02244-f001:**
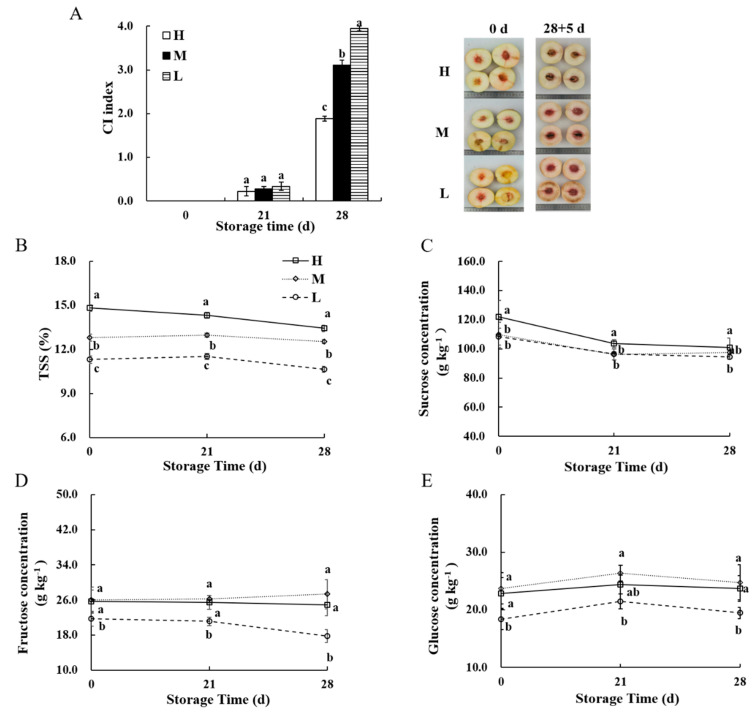
The CI index (**A**), TSS (**B**), sucrose (**C**), fructose (**D**), and glucose (**E**) in peach fruit with different sugar concentrations during storage. Each value is the mean for three replicates, with vertical bars indicating standard errors. The different lower-case letters at each time point indicate a significant difference at *p* ≤ 0.05 by Duncan’s multiple range tests.

**Figure 2 foods-12-02244-f002:**
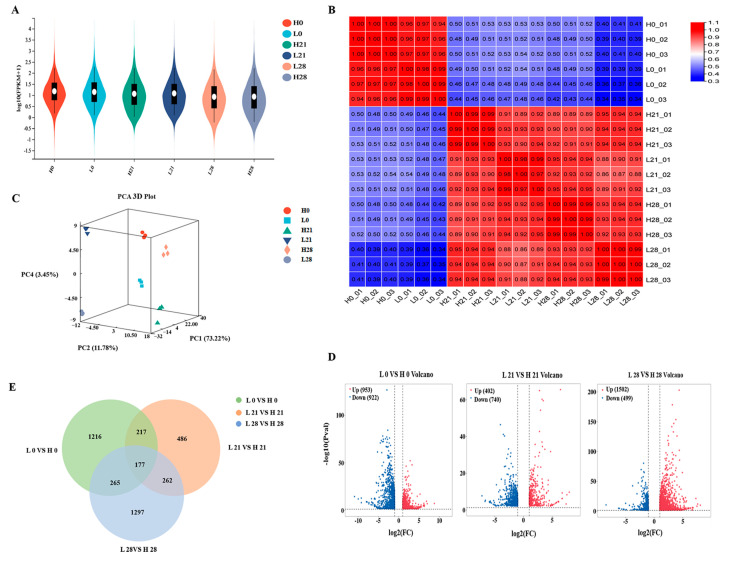
Overview of the transcriptome sequencing of peach fruit with different sugar concentrations under low−temperature storage. A violin plot representation of the expression range from 6 samples (**A**), Pearson correlation between the samples analyses (**B**), PCA analysis among 18 samples (**C**), number of up− and down−regulated genes in L 0 vs. H 0, L 21 vs. H 21, and L 28 vs. H 28 volcano (**D**), and a venn diagram (**E**).

**Figure 3 foods-12-02244-f003:**
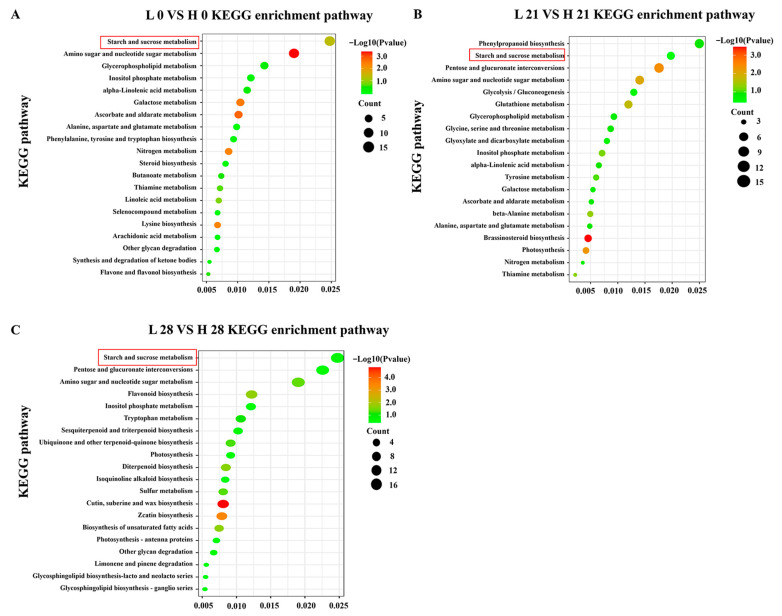
KEGG pathway classification of the DEGs in the three comparison groups of peach fruit. KEGG pathway classification of the DEGs in L 0 vs. H 0 (**A**); KEGG pathway classification of the DEGs in L 21 vs. H 21 (**B**); and KEGG pathway classification of the DEGs in L 28 vs. H 28 (**C**). Starch and sucrose metabolic pathways were mainly analyzed (marked in red).

**Figure 4 foods-12-02244-f004:**
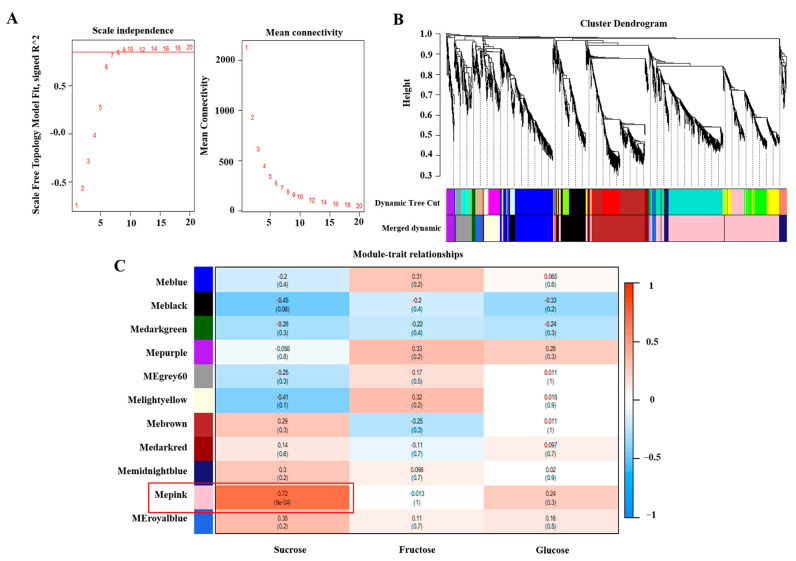
WGCNA of DEGs identified from high-sugar and low-sugar peach fruit. Network topology of different soft−thresholding powers (**A**), hierarchical cluster tree with color annotation for co-expressed genes (**B**), and module-color correlations and corresponding *p*-values, and the picture shows 11 modules, and the right color scale shows the modules’ correlations from −1 to 1 (**C**). The MEpink module was mainly analyzed (marked in red).

**Figure 5 foods-12-02244-f005:**
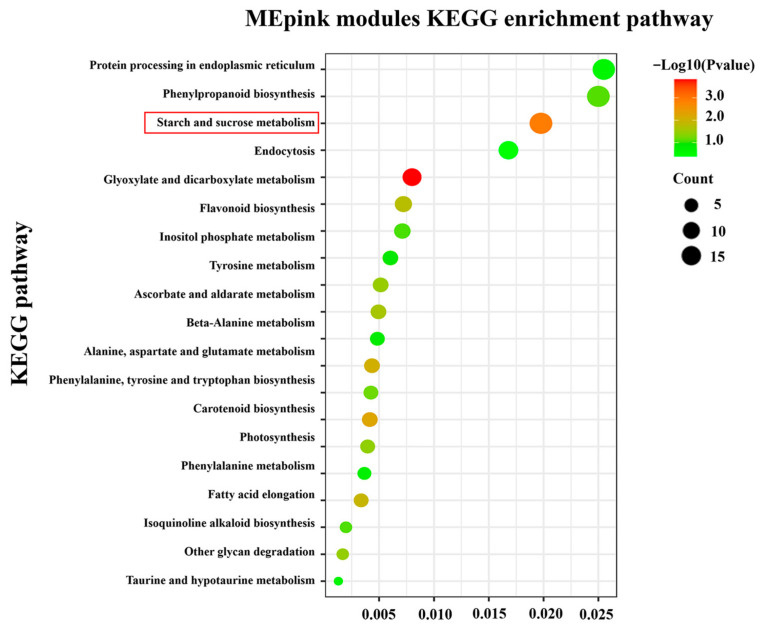
KEGG pathway classification of the 1468 DEGs in the 3 comparison groups of the MEpink module. Starch and sucrose metabolic pathways were mainly analyzed (marked in red).

**Figure 6 foods-12-02244-f006:**
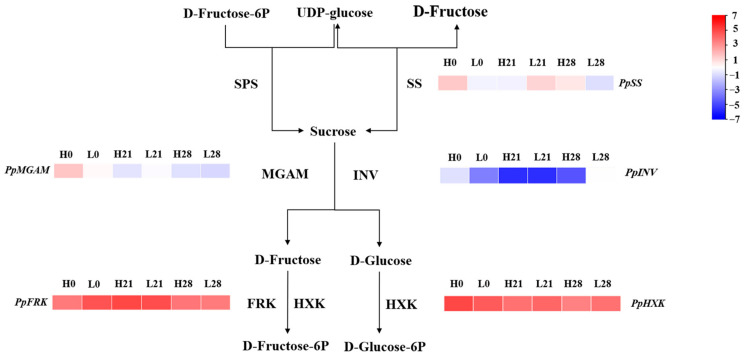
Major DEGs identified in sugar metabolism in the MEpink module during storage. Red or blue colors mean up-regulated or down-regulation, respectively. The value was showed by log_2_ (Fold Change). Each value is the mean for three replicates. Five genes were selected to analyze in this study, including *PpSS* (Prupe.5G241700), *PpMGAM* (Prupe.4G103600), *PpINV* (Prupe.3G009500), *PpHXK* (Prupe.3G057800), and *PpFRK* (Prupe.3G160500).

**Figure 7 foods-12-02244-f007:**
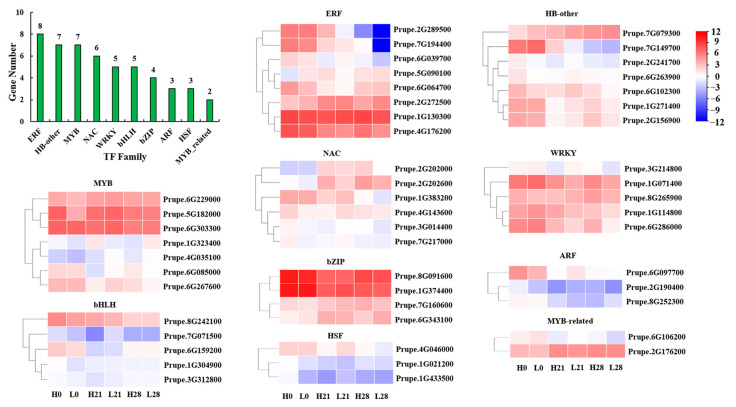
The number and heatmap of differentially expressed TFs of the MEpink module. The figure shows the top ten families of transcription factors, including ERF, HB−other, MYB, NAC, WRKY, bHLH, bZIP, ARF, HSF, and MYB−related, respectively. Red or blue colors mean up-regulated or down-regulation, respectively. The value was showed by log_2_ (Fold Change). Each value is the mean for three replicates. The genes were selected to analyze in this study.

**Figure 8 foods-12-02244-f008:**
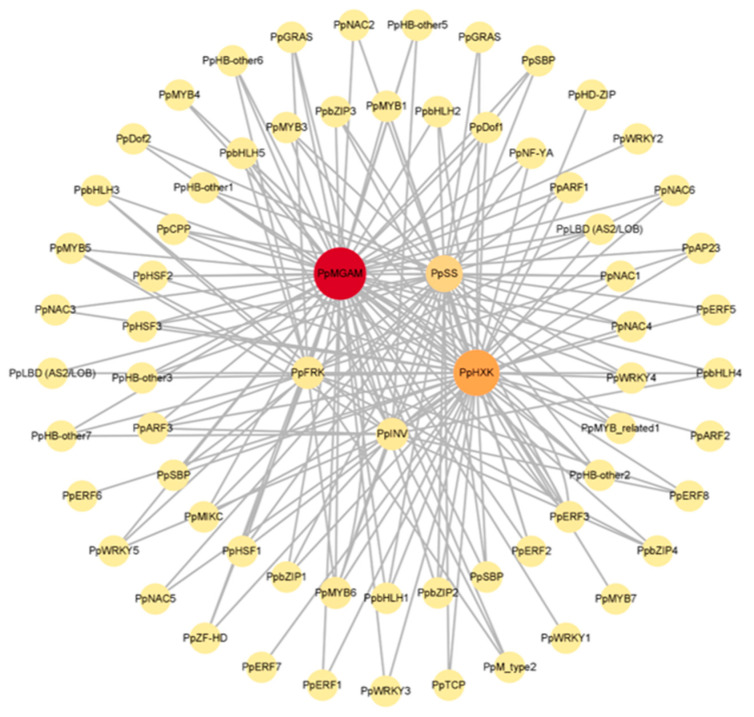
Regulatory network of 81 TFs and 5 functional genes related to sugar metabolism during low temperature stress. The accession number of each gene was listed in Appendix A.

**Figure 9 foods-12-02244-f009:**
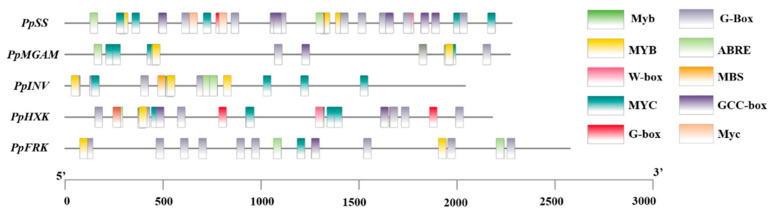
Prediction of cis-acting elements of promoter sequence of 5 functional genes, including *PpSS* (Prupe.5G241700), *PpMGAM* (Prupe.4G103600), *PpINV* (Prupe.3G009500), *PpHXK* (Prupe.3G057800), and *PpFRK* (Prupe.3G160500). The left shows the position of the cis-acting element, and the right show the type of cis-acting element.

**Figure 10 foods-12-02244-f010:**
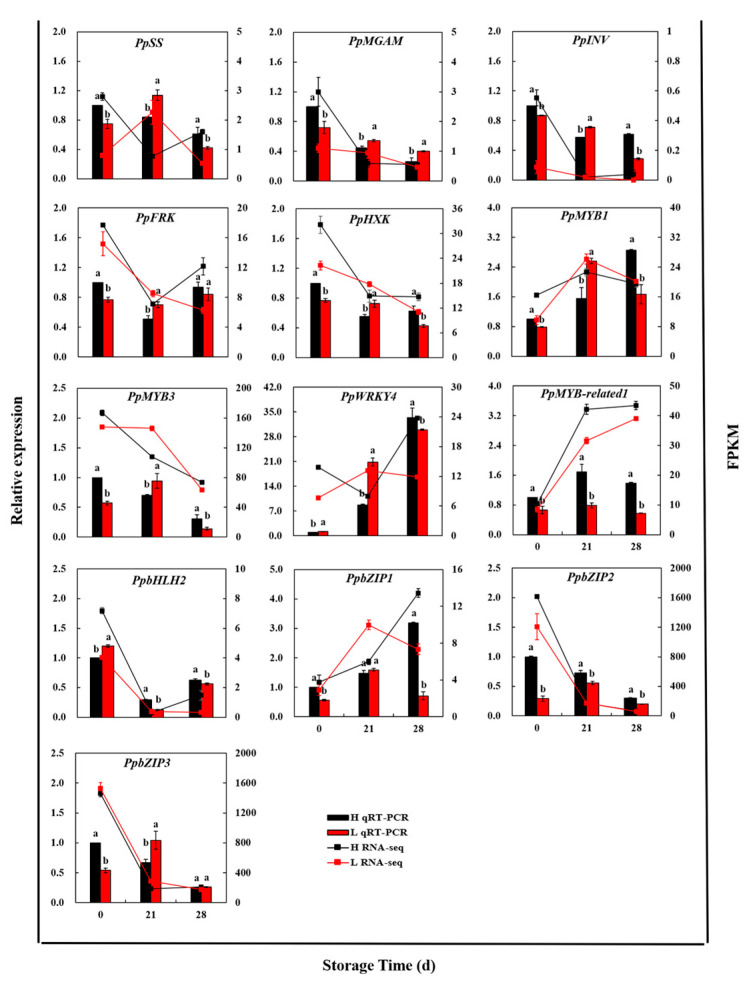
Expression analysis of candidate functional genes and TFs, including *PpSS* (Prupe.5G241700), *PpMGAM* (Prupe.4G103600), *PpINV* (Prupe.3G009500), *PpHXK* (Prupe.3G057800), *PpFRK* (Prupe.3G160500), *PpMYB1* (Prupe.6G229000), *PpMYB3* (Prupe.5G182000), *PpWRKY4* (Prupe.8G265900), *PpMYB-related1* (Prupe.2G176200), *PpbHLH2* (Prupe.6G159200), *PpbZIP1* (Prupe.7G160600), *PpbZIP2* (Prupe.8G091600), and *PpbZIP3* (Prupe.1G374400), as evaluated by real-time PCR. Each value is the mean for three replicates, with vertical bars indicating standard errors. The different lower-case letters at each time point indicate significant difference at *p* ≤ 0.05 by Duncan’s multiple range tests. Expression levels are normalized with respect to *PpTEF2* and are expressed relative to the value of each gene at day 0 of high-sugar peach fruit, which are set to 1.

## Data Availability

Data is contained within the article.

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
