# Peer review of "Transcriptome Co-Expression Network Analysis of Peach Fruit with Different Sugar Concentrations Reveals Key Regulators in Sugar Metabolism Involved in Cold Tolerance"

_foods, 2023, doi:10.3390/foods12112244_

Round 1

Reviewer 1 Report

In the manuscript entitled “Transcriptome co-expression network analysis of peach fruit with different sugar concentrations reveals key regulators in sugar metabolism involved in cold tolerance”, the authors showed the effect of sugar concentration on the low-temperature tolerance of peaches was investigated using transcriptome analysis. It weird to review article without the line number. Nevertheless, I hope my review will be understandable.

The introduction and discussion are too long and unfocused, so it would be helpful to understand the thesis if shortened to only the key parts. Also, references should be written according to the ‘foods’ journal format.

In the materials and methods, how was total RNA extracted? Because fruit contains a large amount of sugar, so RNA extraction is not easy. Please mention whether RNA was extracted from the whole fruit or only selected some tissues. Did you use a special step to remove the sugar?

In results 3.4. Identification of sugar-related DEGs co-expression modules by WGCNA, figure 4 is missing.

In results 3. 9. (figure 10), is the internal reference gene TEF2? please provide the full name. And You should write related reference papers.

From the manuscript, it is not clear whether the original RNA-Seq data has been submitted or not submitted to the NCBI Sequence Read Archive. If they submitted, please provide details and accession numbers.

Minor editing of English language required.

Reviewer 2 Report

The review on the paper by Lufan Wang et al. “Transcriptome co-expression network analysis of peach fruit with different sugar concentrations reveals key regulators in sugar metabolism involved in cold tolerance”

The aim of the study is to better understand the relationship between sugar metabolism and chilling injury (CI) in harvested peach fruits using RNA-Seq approaches. Authors sequenced and analyzed 18 mRNA libraries representing high and low CI resistant fruits at 3 time point after cold treatments/storage conditions. Large amount of DEGs were defined which were functionally annotated and analyzed. Starch and sucrose metabolic pathway was discussed specifically. It was identified five key functional genes and eight TFs that play important roles in regulating sugar metabolism and cold tolerance in peach fruit. Obtained results were confirmed by RT-PCR. Finally, authors claimed that obtained results provide valuable genetic resources for the development of new cold-tolerant peach fruit varieties through molecular breeding.

In general, it is quite simple descriptive paper, mostly stipulated by industrial needs, but which provided interesting results. Methodic was developed with simple “classical” design, based on two contrast samples. The authors did not try to test different temperatures. But despite of these, authors apply quite robust selection of DEGs, which allowed them to select and confirm genes putatively involved in regulating of sugar changes and tolerance to CI.

While the overall high quality of the work, I have several minor essential revisions with this work.

In introduction, it is not very clear what the description was related – to the plant cold tolerance or specifically related to fruits.

Result part is large in general. Some information, e.g. “3.3. Comprehensive analysis of DEGs” could be easily presented as the table

I think all the figures related to GO annotations and maybe KEGG enrichments could be easily moved into supplements and just referred in the text. Because this information does not look to influence the selection of the “Starch and sucrose metabolic pathway” for further analysis.

It is very confusing when used either gene abbreviations (like PpSS) or gene IDs (like Prupe.5G241700) in the figures. It should be uniform.

I am not native English speaker. The language used is quite simple and understandable
